# *Neocinnamomum caudatum* Essential Oil Ameliorates Lipopolysaccharide-Induced Inflammation and Oxidative Stress in RAW 264.7 Cells by Inhibiting NF-κB Activation and ROS Production

**DOI:** 10.3390/molecules27238193

**Published:** 2022-11-24

**Authors:** Sudipta Jena, Asit Ray, Omprakash Mohanta, Prabhat Kumar Das, Ambika Sahoo, Sanghamitra Nayak, Pratap Chandra Panda

**Affiliations:** Centre for Biotechnology, Siksha O Anusandhan (Deemed to be University), Kalinga Nagar, Bhubaneswar 751003, India

**Keywords:** anti-inflammatory, essential oil, lipopolysaccharide, RAW 264.7 cells, *Neocinnamomum caudatum*, NF-κB

## Abstract

*Neocinnamomum caudatum* (Lauraceae) plant is used in the traditional system of medicine and is considered a potential source of edible fruits, spices, flavoring agents and biodiesel. The leaves, bark and roots of the species are used by local communities for the treatment of inflammatory responses, such as allergies, sinusitis and urinary tract infections. However, there is no scientific evidence to support the molecular mechanism through which this plant exerts its anti-inflammatory effect. The aim of the current research was to characterize the chemical constituents of bark (NCB) and leaf (NCL) essential oil of *N. caudatum* and to elucidate its anti-inflammatory action in lipopolysaccharide (LPS)-treated RAW 264.7 cells. Essential oils extracted by hydrodistillation were further subjected to gas chromatography mass spectrometry (GC-MS) analysis. The major constituents in bark essential oil identified as β-pinene (13.11%), *α*-cadinol (11.18%) and *α*-pinene (10.99%), whereas leaf essential oil was found to be rich in β-pinene (45.21%), myrcene (9.97%) and α-pinene (9.27%). Treatment with NCB and NCL at a concentration of 25 µg/mL exerted significant anti-inflammatory activity by significantly reducing LPS-triggered nitric oxide (NO) production to 45.86% and 61.64%, respectively, compared to the LPS-treated group. In the LPS-treated group, the production of proinflammatory cytokines, such as tumor necrosis factor (TNF)-α, interleukin (IL)-6 and IL-1β, decreased after treatment with essential oil, alleviating the mRNA levels of inducible nitric oxide synthase (iNOS) and cyclooxygenase (COX)-2. The essential oil also inhibited the production of intracellular ROS and attenuated the depletion of mitochondrial membrane potential in a concentration-dependent manner. Pretreatment with NCB also reduced nuclear factor kappa-B (NF-κB)/p65 translocation and elevated the levels of endogenous antioxidant enzymes in LPS-induced macrophages. The present findings, for the first time, demonstrate the anti-inflammatory potential of both bark and leaf essential oils of *N. caudatum*. The bark essential oil exhibited a significantly more important anti-inflammatory effect than the leaf essential oil and could be used as a potential therapeutic agent for the treatment of inflammatory diseases.

## 1. Introduction

Inflammation is a complex response against pathogen invasion, tissue damage, chemical irritation and exposure to endotoxins, such as lipopolysaccharide (LPS), that maintain homeostasis in the human body [1,2]. The balance between proinflammatory and anti-inflammatory cytokines is maintained by normal inflammatory responses [3]. However, excessive production of proinflammatory cytokines has been linked to the emergence of chronic inflammatory diseases, such as cancer, inflammatory bowel disease (IBD) and metabolic syndromes [4,5,6].

Macrophages, a major component of the mononuclear phagocyte system, are distributed ubiquitously in tissues, and they support homeostasis and host defense against intracellular parasitic bacteria and pathogenic protozoa [7,8]. LPS is a common inducer from *Escherichia coli* O111:B4 [9,10] that stimulates macrophage activation and upregulates a number of cytokines in RAW 264.7 cells during the inflammatory reaction, including IL-1, TNF-α, IL-6, etc. [11]. The pathogenesis of various inflammatory disorders, such as rheumatoid arthritis, atherosclerosis, chronic hepatitis and pulmonary fibrosis, is caused by excessive production of proinflammatory cytokines [12]. Thus, LPS-treated murine macrophages are commonly used to understand the mechanism of action of possible anti-inflammatory mediators that are involved in the inflammation process [13].

More than 30 million individuals worldwide use nonsteroidal anti-inflammatory drugs (NSAIDs), making them one of the most frequently prescribed medicines for the treatment of inflammation [14]. Unwanted clinical effects linked with the use of NSAIDs are most commonly observed in the gastrointestinal system and the kidneys [15]. Every year, approximately 2.5 million Americans are affected by NSAID-mediated renal consequences [16]. The majority of NSAIDs are anti-inflammatory drugs of the first choice. On the other hand, the regular usage of NSAIDs has been linked to gastrointestinal ulcers, renal damage and kidney failure [17,18]. The search for safe and efficient anti-inflammatory compounds in plants and products derived from plants is becoming increasingly popular, owing to their efficacy and comparatively few or no side effects. Therefore, researchers have shifted their attention toward plant-based therapies for the treatment of a variety of inflammatory disorders.

The genus *Neocinnamomum* (Lauraceae) is comprised of six species, which are endemic to tropical Asia [19,20,21]. The leaves of *N. delavayi* are used for the treatment of rheumatic disorders in China [21]. Varies parts of *Neocinnamomum caudatum*, which is widespread in the subtropical regions of South China, India and Myanmar, are utilized in the traditional Chinese and Indian systems of medicine [22,23,24,25]. In India, the leaves, bark and roots are used to cure inflammatory responses, such as allergies, sinusitis, urinary tract infection and injuries [23,25]. The fruits of this plant are consumed in India [26,27,28]. The bark of the plant is also used as a spice and flavoring agent in India and elsewhere [23,29,30]. *N. caudatum* has recently attracted attention for its seed oil content and unique fatty acid composition and has been suggested as an alternative source of biodiesel in China [24,31].

Although no reports have been published on the anti-inflammatory activities of essential oil of any *Neocinnamomum* species to date, the essential oils of several species of its closely related genus, *Cinnamomum*, are potent anti-inflammatory agents [32,33,34,35,36,37,38,39,40,41,42]. Therefore, it is expected that the essential of oil *Neocinnamomum* species might have similar anti-inflammatory effects. Furthermore, GC -MS analysis of the essential oil of both leaf and bark of *Neocinnamomum caudatum* performed in the present study reveals the predominance of α-pinene, β-pinene and myrcene, which are known for their anti-inflammatory activities [43,44,45,46,47,48,49,50,51,52,53,54,55]. This prompted us to investigate the anti-inflammatory effect of the essential oil of *N. caudatum*. 

Despite the therapeutic value of *N. caudatum*, especially for treatment of inflammatory responses, no reports have been published on the anti-inflammatory activity of its essential oil. Thus, the objective of the present study was to determine the phytochemical composition of leaf and bark essential oil of *N. caudatum* and to evaluate its anti-inflammatory activity in lipopolysaccharide (LPS)-treated murine macrophages.

## 2. Results and Discussion

### 2.1. Volatile Profiling of N. caudatum Essential Oil

The leaf (NCL) and bark (NCB) essential oils of *N. caudatum* were pale yellowish in color, with a yield of 0.5 and 1.2 (%*v*/*w*), respectively, on a dry-weight basis, which was higher than the yield percentage of leaf essential oil of *N. delavayi* (0.80%) [56]. A total of 45 and 41 constituents accounting for 95.55 and 98.93% of the total leaf (NCL) and bark essential oil (NCB), respectively, were identified by GC-MS (Table 1). The predominant constituents in the NCL essential oil were β-pinene (45.21%), myrcene (9.97%), α-pinene (9.27%), α-terpineol (3.95%) and 1,8-cineole (2.66%). β-pinene (13.11%) was the dominant constituent in the NCB essential oil, followed by α-cadinol (11.18%), α-pinene (10.99%), myrcene (7.81%) and α-terpineol (6.79%). The NCL essential oil was characterized by chemical groups such as monoterpene hydrocarbons (75.24%), sesquiterpene hydrocarbon (10.68%), monoterpene alcohol (4.65%), monoterpene ether (2.66%), monoterpene ester (1.64%), sesquiterpene alcohol (0.54%) and monoterpene aldehyde (0.14%). Similarly, the NCB essential oil was rich in monoterpene hydrocarbons (42.73%), sesquiterpene hydrocarbon (24.42%), sesquiterpene alcohol (22.18%) and monoterpene alcohol (8.62%). Although the leaf essential oil of *Neocinnamomum delavayi* from China was reported to contain α-phellandrene (47.55%) as the predominant constituent, α-pinene (17.35%) and β-pinene (9.77%) were also present in considerably higher amounts [56]. In the related genus, *Cinnamomum*, although the essential oils isolated from most of the species contain cinnamaldehyde and its derivatives as the major constituents, a higher percentage of α-pinene and β-pinene has also been reported in species such as *C. glanduliferum* [57], *C. tonduzi* [58] and a few others.

Monoterpene hydrocarbons were the major class of compounds in *N. caudatum* essential oil, followed by sesquiterpene hydrocarbons. The essential oils of *Liquidambar styraciflua*, *Citrus unshiu* and species of *Laserpitium* containing α-pinene, a monoterpene, have been reported to influence inflammation-related conditions [59,60,61]. In particular, α-pinene was found to be useful in modulating bone resorption, acute pancreatitis and allergic rhinitis [45,62,63]. β-pinene was found to be 2 to 12 times more effective than α-pinene against bacterial and fungal diseases [64], whereas α-pinene has been reported to exert an anti-inflammatory effect in mice via suppression of MAPKs and the NF-κB pathway [60], and *β*-pinene has been shown to reduce carrageenan-induced paw edema and leukocyte migration in rats [65]. Very strong 5-lipoxygenase inhibitory activities of the essential oil of *Salvia* species have been attributed to the presence of α-pinene [66], and the leaf essential oil of *Alpinia* species is considered an effective 5-lipoxygenase inhibitor, owing to high contents of both β-pinene and α-pinene [67]. Likewise, myrcene, another predominant component found in *N. caudatum* essential oil, was found to suppress the inflammatory response generated by lipopolysaccharide (LPS) in mice by inhibiting cell migration and release of nitric oxide [43]. 

### 2.2. Effect of N. caudatum Essential Oil on Cell Viability and NO Production of RAW 264.7 Cells

The cytotoxic effect of NCB and NCL essential oil on RAW 264.7 cells was analyzed through MTT assay (Figure 1). The cells were exposed to various concentrations of leaf and bark essential oil (6.25, 12.5, 25, 50 and 100 μg/mL) for 24 h, but no cytotoxicity was observed in RAW 264.7 cells at all the tested concentrations. The viability of the cells decreased with an increase in the concentrations of *N. caudatum* essential oils from 6.25 to 100 µg/mL. Therefore, these two concentrations (12.5 and 25 μg/mL) were selected for in vitro anti-inflammatory study. To investigate the anti-inflammatory activity of the leaf essential oil of *Cinnamomum osmophloeum*, NO production in LPS-stimulated RAW 264.7 cells was estimated by Tung et al. [68]. They reported excellent inhibitory activity on NO production, with IC_50_ values ranging from 9.7 to 15.5 μg/mL. Similarly, treatment with leaf essential oil of *Cinnamomum glanduliferum* at doses of 250, 500 and 1000 mg/kg was found to significantly modulate ethanol-induced gastritis in rats as the level of NO reduced to 32, 37 and 41 µM nitrate/g [35]. Fruit essential oil of *Cinnamomum insularimontanum* also exhibited significant NO production inhibitory activity, with an IC_50_ value of 18.68 µg/mL [33].

The effect of NCB and NCL essential oil on the level of nitrite production in RAW 264.7 cells was determined by a Griess reagent. In RAW 264.7 cells, no significant difference was observed in nitrite production following treatment with *N. caudatum* essential oils as compared to the control (Figure 2). However, nitrite accumulation increased by 92.76% (13.8-fold) after incubation with LPS in comparison to the control group, whereas treatment with NCB essential oil at doses of 12.5 and 25 µg/mL significantly decreased LPS-induced nitrite production by 37.5% (1.6-fold) and 54.55% (2.2-fold), respectively. Nitric oxide is a crucial inflammatory mediator that regulates a variety of pathological states within the cell [69]. Under physiological conditions, nitric oxide plays an important role in various processes, such as the host pathogen defense, neuron communication and vasodilation [70]. 

However, the overproduction of nitric oxide leads to acute and chronic inflammation [71]. Therefore, inhibition of NO generation in murine macrophages is a possible strategy to reduce the severity of tissue damage in cases of inflammatory disorders. The result of the present study demonstrate that NCB and NCL essential oil can exert anti-inflammatory effects by reducing the levels of nitric oxide in RAW 264.7 cells.

### 2.3. Effect of N. caudatum Essential Oil on Proinflammatory Cytokines

The effect of treatment with *N. caudatum* essential oil on the production of proinflammatory cytokines in the LPS-induced RAW 264.7 cells was determined by ELISA. The levels of IL-6, IL-1β and TNF-α were significantly elevated from 1.004 to 5.495 pg/mL, 1.001 to 12.730 pg/mL and 1.003 to 7.988 pg/mL, respectively, after 24 h treatment with LPS alone (Figure 3). 

Pretreatment with 25 µg/mL bark essential oil resulted in a concentration of IL-6 and IL-1β of 1.172 and 1.110 pg/mL, respectively, as compared to the LPS group. However, pretreatment with bark essential oil at a concentration of 25 µg/mL significantly reduced the level of TNF-α by 2.8-fold as compared to the LPS-treated group. 

The results suggest that both leaf and bark essential oil of *N. caudatum* brought about a concentration-dependent reduction in the level of proinflammatory cytokines, such as IL-6, IL-1β and TNF-α. The findings of the current work indicate that leaf and bark essential oil of *N. caudatum* can effectively inhibit the expression levels of proinflammatory cytokines in RAW 264.7 cells induced by LPS. Proinflammatory cytokines, including TNF- α, IL-1β and IL-6, are known to be released by murine macrophages during inflammatory response [72], and these proinflammatory cytokines play a vital role in inflammation, including activation of acute-phase response and leukocytes [73]. The anti-inflammatory effects of the leaf essential oil of *Cinnamomum subavenium* were evaluated by LPS-stimulated RAW264.7 cells, and the levels of proinflammatory mediators, such as TNF-𝛼, IL-1𝛽 and IL-6, which were considerably increased by LPS, were significantly decreased by the essential oil [37]. Similar findings were reported in the case of *Cinnamomum burmannii* bark oil [40] and leaf oil of *Cinnamomum cassia* [42]. The findings of the present investigation are consistent with those of similar studies performed on RAW 264.7 cells, wherein plant essential oils were reported to suppress the production of proinflammatory cytokines generated by LPS [13,74,75]. 

### 2.4. Effect of N. caudatum Essential Oil on LPS-Induced iNOS and COX-2 mRNA Expression

To assess the anti-inflammatory activity of NCB and NCL essential oil, the production of iNOS and COX-2 in LPS-induced cells was estimated by qRT-PCR. RAW 264.7 cells were treated with varying concentrations of NCB and NCL essential oils (12.5 and 25 µg/mL), as well as LPS, and incubated for 24 h. After incubation, the mRNA expression level of iNOS and COX-2 in RAW 264.7 cells was measured. The mRNA expression levels of COX-2 (Figure 4A) and iNOS enzyme (Figure 4B) showed a concentration-dependent reduction in concentration in the RAW 264.7 cells treated with NCB and NCL essential oils at doses of 12.5 and 25 µg/mL. Interestingly, the RAW 264.7 cells treated with the higher dose of bark essential oil showed decreased COX-2 expression as compared to the LPS-treated group. iNOS is the most common isoform of Nitric oxide (NO) synthase found in murine macrophages [76]. LPS, a component of the cell walls of Gram-negative bacteria, is a potent activator of iNOS production in macrophages [77]. LPS interacts with Toll-like receptor 4 and activates macrophages [3]. The COX-2 enzyme is involved in the synthesis of prostaglandin E2 (PGE2), an essential inflammatory mediator. According to Wei et al. [78], PGE2 production is closely linked to NO production. 

The reduction in iNOS and COX-2 expression is considered a crucial event in reducing inflammation. The present findings suggested that the leaf and bark essential oil of *N. caudatum* exhibited anti-inflammatory activity by downregulating iNOS and COX-2 mRNA expression in RAW 264.7 cells. Activated NF-𝜅B regulates the transcription of inflammatory cytokines and enzymes, such as iNOS and COX-2 [79]; therefore, the NF-𝜅B signaling pathway is considered a therapeutic route against inflammatory diseases. A significant decrease in iNOS and COX-2 immunoreactive cells of paw tissue treated with leaf oil (200 mg/kg) of *Cinnamomum subavenium* was observed [37]. The authors concluded that the essential oil possesses anti-inflammatory properties, and the effects might have been caused by inhibition of iNOS and COX-2 expression, affecting the NF-𝜅B pathway.

### 2.5. Effect of N. caudatum Essential Oil on LPS-Induced Reactive Oxygen Species (ROS) Levels 

Oxidative stress plays a pivotal role in the pathogenesis of inflammatory diseases, such as cancer and cardiovascular disorders [80,81]. Multiple molecular targets linked to acute and chronic inflammation, growth differentiation, proliferation and apoptosis are modulated by ROS [82].

The immunological response to inflammation generates reactive oxygen species (ROS), disrupting the body’s normal oxidant/antioxidant balance and damaging cellular organelles [83]. LPS stimulates the generation of reactive oxygen species (ROS) in macrophages, which activates the NF-κB inflammatory signaling pathway and destroys mitochondrial membrane integrity [84].

In the current investigation, the amount of ROS expression in RAW 264.7 cells was determined using an oxidation-sensitive probe i.e., 2′, 7′-dichlorodihydrofluorescein diacetates (DCFH-DA), to determine whether *N. caudatum* essential oil had an effect on ROS accumulation. Fluorescence microscopic images and fluorescence intensity measurements revealed that increased ROS accumulation by LPS was attenuated by pretreatment of the RAW 264.7 cells with NCB and NCL essential oils at a dose of 25 µg/mL (Figure 5). However, pretreatment with NCL and NCB essential oils at a dose of 25 µg/mL significantly reduced the intracellular ROS by 90.90% (11-fold) and 87.95% (8.3-fold), respectively, as compared to the LPS-treated group. The intracellular level of ROS in LPS-induced RAW 264.7 cells was dramatically reduced after treatment with *N. caudatum* essential oil. The reduced NF-κB activity and enhanced mitochondrial membrane potential could explain the reduced levels of ROS in RAW 264.7 cells treated with *N. caudatum* essential oil.

### 2.6. Effect of N. caudatum Essential Oil on LPS-Induced Antioxidant Enzymes in RAW 264.7 Cells

One of the key indicators of oxidative stress in the body is the ratio of reduced to oxidized glutathione within cells. Increased levels of ROS in the body have been linked to altered DNA structure, modified proteins and lipids, stimulation of several stress-regulated transcription factors and the production of inflammatory cytokines [85]. The protective effects of antioxidant enzymes are impeded by the excessive production of reactive oxygen species [86]. Phytocompounds derived from plants have been shown, in several studies, to exhibit a protective effect against oxidative stress by enhancing endogenous antioxidant enzymatic activity and re-establishing the normal condition of the cells. Therefore, we measured the endogenous antioxidant enzyme levels in RAW 264.7 cells treated with *N. caudatum* essential oil. SOD, CAT, GPx and GSH activities were measured with the treatment of *N. caudatum* essential oils compared to the LPS group in RAW 264.7 cells. Pretreatment with NCB essential oil at doses of 12.5 and 25 µg/mL significantly increased SOD levels by 1.6- and 2.9-fold, respectively, as compared to the LPS-treated group.

Similarly, NCL essential oils at doses of 12.5 and 25 µg/mL also resulted in a 1.4- and 2.7-fold increase in SOD levels, respectively, as compared to the LPS-treated group (Figure 6A–D). In comparison to LPS treatment, the pretreatment with NCB essential oil at doses of 12.5 and 25 µg/mL enhanced the concentration of CAT by 1.3- and 3.7-fold, GSH by 7- and 3-fold and GPx by 1.7- and 6.7-fold, respectively. Similarly, treatment with NCL essential oil at doses of 12.5 and 25 µg/mL led to a significant increase in the level of CAT by 1.3- and 3.3-fold, GSH by 2.7- and 6.8-fold and GPx by 1.07- and 6.5-fold, respectively. Superoxide dismutase (SOD) is known to remove superoxide from living organisms and protect them from ROS-induced injury, and catalase mediates its action by removing hydrogen peroxide produced by lipid auto-oxidation and organic compound oxidation. In the current study, we established that *N. caudatum* essential oil can attenuate LPS-induced oxidative stress by enhancing the activities of antioxidative enzymes, such as catalase, SOD, GPx and GSH. 

### 2.7. Effect of N. caudatum Essential Oil on Mitochondrial Membrane Potential

Increased reactive oxygen species destroy the mitochondrial membrane integrity when RAW 264.7 cells are induced with LPS. In order to analyze the effect of *N. caudatum* essential oil on LPS-induced mitochondrial damage, we treated RAW 264.7 cells with the mitochondrial specific probe JC-1 and visualized them under a fluorescent microscope (Figure 7). When the membrane potential was normal, JC-1 fluoresced red, and when the membrane potential was disrupted, it fluoresced green. This loss of membrane potential was significantly restored by treatment with *N. caudatum* essential oil at concentrations of 12.5 and 25 μg/mL.

### 2.8. Effect of N. caudatum Essential Oil on LPS-Induced NF-κB Activation 

The NF-κB transcription factor is involved in various cellular processes, such as proliferation, cell death and immunological responses. NF-κB is generally sequestered in the cytoplasm by an inhibitory protein class known as the inhibitor of κB (IκB) [87]. NF-κB is a heterodimer complex consisting of p65 and p50 subunits. NF-κB, together with its inhibitory protein, IκBκ, is found in the cytoplasm. Following LPS stimulation, NF-κB is liberated from its inhibitory protein, IκBκ, which is then phosphorylated, ubiquitinated and degraded by the proteasome. Then, NF-κB is translocated to the nucleus, thereby promoting the expression of inflammatory genes and cytokines/mediators [88,89]. To assess the role of *N. caudatum* essential oil in inhibiting the nuclear translocation of NF-κB from cytoplasm, RAW 264.7 cells were exposed to *N. caudatum* essential oil for 24 h, followed by treatment with LPS. Confocal microscopy was used to examine the translocation of NF-κB p65 to the nucleus (Figure 8). After treatment, the RAW 264.7 cells were immunostained against the NF-κB p65 antibody and incubated for 12 h. The nucleus was stained with a fluorescent stain (Hoechst 33342) and FITC-conjugated secondary antibody. LPS stimulation increased NF-κB p65 levels in the nucleus, whereas pretreatment of the cells with NCB essential oil at doses of 12.5 and 25 μg/mL inhibited the nuclear translocation of NF-κB by 72.97% (3.7-fold) and 92.31% (13-fold), respectively, as compared to LPS-treated cells. Based on experiments on the effect of *Cinnamomum subavenium* essential oil using a carrageenan-induced hind mouse paw oedema model [37], it was established that NF-𝜅Bp65, a signal for NF-𝜅B activation, played an important role in the regulation of its transcriptional capacity, and the essential oil suppressed the nuclear translocation of NF-𝜅Bp65 in a concentration-dependent manner. 

### 2.9. Differential Anti-Inflammatory Activity of Bark and Leaf Essential Oil of N. caudatum

The relatively superior anti-inflammatory activity of *N. caudatum* bark (NCB) found in the present study might be due to the presence of compounds such as α-terpineol and α-cadinol in higher amounts in NCB as compared to NCL. Several reports have shown that α-cadinol and α-terpineol exhibit significant anti-inflammatory activities [68,90]. α-terpineol displayed anti-inflammatory properties by inhibiting the cytokine cascade generated by carrageenan and inhibiting the production of nitric oxide in mice. Compound α-cadinol showed potential anti-inflammatory properties by inhibiting NO production in LPS-induced macrophages [68]. Sabinene, a monoterpene hydrocarbon identified in the bark essential oil of *N. caudatum*, is known to exhibit significant anti-inflammatory effects through the inhibition of proinflammatory cytokines, such as IL-1β, IL-6 and TNF-α [91]. Thus, the significant anti-inflammatory activity of NCB essential oil might be directly related to the scavenging ability of the compounds present in it or their capacity to inhibit iNOS and COX-2 expression, the enzyme responsible for the release of high amounts of NO under inflammatory conditions. Additionally, the anti-inflammatory effect of NCB might be due to the synergistic interaction of major and minor constituents present in the essential oil. 

## 3. Materials and Methods 

### 3.1. Essential Oil Extraction

Fresh leaves and bark of *Neocinnamomum caudatum* were collected from Deomali Hill, Koraput district, Odisha (83°0’22.9068” E, 18°38’33.9612” N, alt. 1122.03 msl) in January 2021, and the identities were confirmed by Prof. P. C. Panda, taxonomist. The herbarium specimen (2258/CBT Dt. 15.1.2021) was kept at the Herbarium of CBT, SOA University, Bhubaneswar, as a voucher. The essential oil was isolated from the dried leaves and bark of *N. caudatum* (500 g each) by hydrodistillation according to the method reported in the European Pharmacopoeia [92]. The hydrodistillation process was repeated thrice. Further, anhydrous Na_2_SO_4_ was used to remove traces of water in the extracted essential oil, and the oil was stored at 4 °C. 

### 3.2. Chemical Characterization of Essential Oil

Individual samples were analyzed on a Clarus 580 gas chromatograph equipped with a SQ-8 MS detector. Volatile constituents of essential oils were separated on an Elite-5 MS capillary column (30 m × 0.25 mm × 0.25 µm) with an electron ionization source set at 70 eV. The flow rate of the helium was set at 1 mL/min, and 0.1 µL of the undiluted essential oil was injected. The oven temperature program was set at 60 °C, increased to 220 °C at 3 °C/min and then maintained at this temperature for 7 min. The individual compound was identified by comparing the mass spectra of detected constituents with in-house NIST library spectra. Further identification was also performed by coinjecting authentic standards and by comparison of the RI with values reported in the literature [93].

### 3.3. Cytotoxicity Assay 

RAW 264.7 cells were maintained in DMEM supplemented with FBS, 2 mM L-glutamine and penicillin-streptomycin (100 µg/mL). The cells were grown in a humidified incubator at 37 °C. Varying concentrations of essential oils were prepared by dissolving them in DMSO and freshly dilution to the desired concentration with the culture medium. The final concentration of DMSO was 0.5% (*v/v*). The cytotoxic activity of the essential oils and positive control was assessed according to a modified method reported Ray et al. [94]. The cells (1 × 10^5^ cells/mL) were plated in 96-well plates in the presence of various concentrations of essential oils (6.25, 12.5, 25, 50 and 100 μg/mL) for 24 h in a humidified atmosphere. After incubation, MTT reagent (5 mg/mL) was added and kept for 3 h. DMSO (100 µL) was added after removing the MTT reagent to solubilize the formazan crystals generated during the reaction. The absorbance of the solution was then measured using an ELISA plate reader at 540 nm. 

### 3.4. Determination of Nitric Oxide Production

The RAW 264.7 cells were treated with varying concentrations of essential oils (12.5 and 25 µg/mL), followed by treatment with LPS (1 µg/mL) at 37 °C for 24 h. Subsequently, the level of NO production in the culture medium was measured using a Griess reagent [95]. Then, Griess reagent (100 μL) was added to the culture medium (100 μL) and incubated for 10 min at 37 °C, followed by measurement of optical density at 540 nm.

### 3.5. Estimation of Proinflammatory Cytokines 

The levels of proinflammatory cytokines were determined using immunoassay ELISA kits according to the supplier’s protocol (Abcam Co., Cambridge, UK). The RAW 264.7 cells were incubated with leaf and bark essential oil of *N. caudatum* (12.5–25 μg/mL), followed by treatment with LPS (1 μg/mL). After 24 h incubation, the culture medium from each well was collected and centrifuged. After centrifugation, supernatants were harvested for estimation of IL-6, IL-1β and TNF-α. The concentrations of proinflammatory cytokines in the sample were evaluated based on the standard curve. All experiments were carried out in triplicate.

### 3.6. mRNA Expression of iNOS and COX-2 

The mRNA expression levels of iNOS and COX-2 in RAW 264.7 cells were estimated by quantitative real-time PCR (qRT-PCR) according to the protocol reported by Syam et al. [96]. The RAW 264.7 cells were plated in a culture dish at a density of 1 × 10^5^ cells/mL and incubated until they reached confluency. The culture medium was then replaced with serum-free medium containing varying concentrations of NCB and NCL essential oil and LPS. Then, the plate was cultivated for another 24 h under the same conditions. 

RNA was isolated using a Qiagen RNeasy kit (Hilden, Germany). RNA was reverse-transcribed into cDNA using an IScript cDNA synthesis kit (Biorad). PCR amplification was performed using gene-specific forward and reverse primers: COX-2, forward (5′-GGA GAG ACT ATC AAG ATA GT-3′) and reverse (5′-ATG GTC AGT AGA CTT TTACA-3′); iNOS, forward (5′- AGA CTG GAT TTG GCT GGT CCC TCC-3′) and reverse (5′-AGA ACT GAG GGT ACA TGC TGG AGCC-3′); GAPDH, forward (5′-CGG AGT CAA CGG ATT TGG TCG TAT-3′) and reverse (5′-GCA GGT CAG GTC CAC CAC TGA C-3′). 

PCR analysis was performed with a Qiagen Rotor Gene Q 5plex HRM using SYBR green fluorescent dye under the following conditions: 95 °C for 5 min, then 40 cycles at 95 °C for 5 s, 60 °C for 30 s and a final extension at 72 °C for 20 s. The cycle threshold (Ct) was detected, and the average Ct values were calculated. The relative expression of each gene was measured and normalized using the 2^−∆∆Ct^ method relative to the housekeeping gene (GADPH) of the same sample. 

### 3.7. Estimation of Reactive Oxygen Species (ROS) Levels

2′, 7′-dichlorodihydrofluorescein diacetate (DCFH-DA) was used to estimate the level of intracellular ROS in RAW 264.7 cells. Cells (1 × 10^5^ cells/mL) were plated in a 24-well plate and treated with varying concentrations of leaf and bark essential oils (12.5 and 25 µg/mL), followed by LPS (1 µg/mL) for 24 h. Subsequently, DCFH-DA (5 µM) was added and incubated for 30 min. Then, the culture medium was discarded, and the wells were washed twice using cold PBS. The cell pellets were then dissolved in PBS, and intracellular ROS levels were observed under a confocal microscope (Zeiss LSM 880, Jena, Germany).

### 3.8. Estimation of Endogenous Antioxidant Enzymes

RAW 264.7 cells were incubated with various concentrations of NCB and NCL essential oil and LPS (1 µg/mL) for 24 h. The harvested cells were lysed and centrifuged to obtain a cell lysate. Various endogenous enzymes, such as GSH, CAT, SOD and GPx, were estimated from the cellular lysate according to the supplier’s protocol (Oxis Research, Portland, USA). The results of GSH are expressed as µM, SOD as U/mg, and GPx and CAT as ng/mL protein and mU/mL, respectively. 

### 3.9. Estimation of Mitochondrial Membrane Potential

The mitochondrial membrane potential (ΔΨm) was determined by JC-1 dye (mitochondrial membrane potential dye) as previously described by Ray et al. [60]. Briefly, the RAW 264.7 cells were seeded at a concentration of 1 × 10^5^ cells/mL in culture plates and grown overnight. The cells were subsequently incubated for 24 h with various concentrations of essential oils (12.5 and 25 µg/mL) and LPS (1 µg/mL). The cells were washed with PBS and stained with JC-1 dye (1 µg/mL) for 30 min in the dark at 37˚C. Then, the images were observed under a confocal microscope (Zeiss LSM 880, Germany). Quantitative analysis of the ratio of red/green fluorescence in each group was conducted using Image J software 

### 3.10. Assessment of NF-κB Nuclear Translocation 

RAW 264.7 cells were grown in a culture plate at a seeding density of 1 × 10^5^ cells/mL under humidified conditions. The cells were then treated with various concentrations of leaf and bark essential oil (12.5 and 25 µg/mL), followed by LPS (1 µg/mL) for 24 h. The culture medium was removed from all wells and rinsed with PBS at the end of the treatment. Furthermore, BD Cytofix/Cytoperm solution (0.5 mL) was mixed and kept for 10 min. Then, the cells were washed with 0.5% bovine serum albumin (BSA). The cells were stained with 10 µL of PE mouse anti-NFkB p65 antibody for 30 min and counter-stained with 100 µL of DAPI solution (1 µg/mL) for 10 min in the dark before imaging. The cells were observed under a confocal microscope (Zeiss LSM 880, Germany), and Image J software was used to measure the expression of NFkB-p65.

### 3.11. Statistical Analysis

Data obtained from the analysis are represented as the mean ± SD of three independent experiments. An ANOVA was performed, followed by Tukey’s multiple range test, using GraphPad Prism software.

## 4. Conclusions

The present findings show that the leaf and bark essential oil of *Neocinnamomum caudatum* can inhibit LPS-induced inflammatory and oxidative stress in RAW 264.7 cells, as evidenced by a decrease in the levels of proinflammatory cytokines, inflammatory mediators and intracellular ROS. The essential oil of this species can also alleviate the depletion of mitochondrial membrane potential and decrease NF-κB nuclear translocation in LPS-induced RAW 264.7 cells. The results of the current work indicate that *N. caudatum* essential oil might be used as an alternate source of anti-inflammatory agents for the prevention and reversal of anti-inflammatory responses. However, further studies may be necessary to establish the signaling pathways responsible for the anti-inflammatory effects of its essential oil.

## Figures and Tables

**Figure 1 molecules-27-08193-f001:**
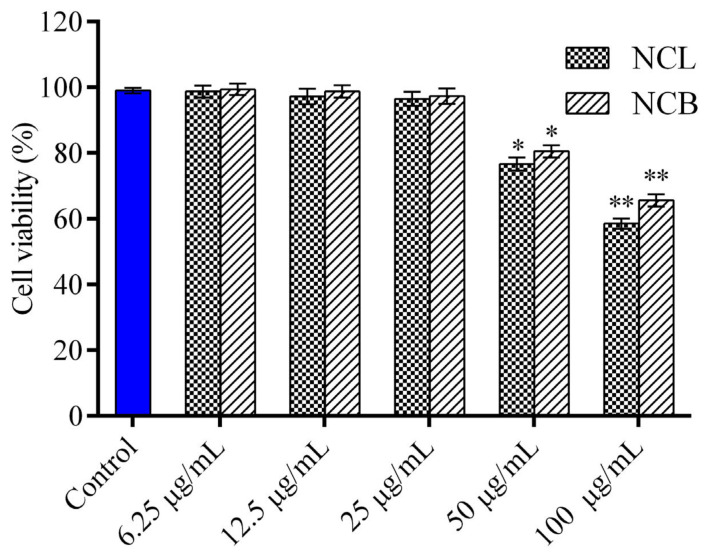
Effect of *N. caudatum* leaf and bark essential oils on cell viability. RAW 264.7 cells were treated with various concentrations of *N. caudatum* leaf and bark essential oils (6.25–100 µg/mL) for 24 h. The total number of viable cells was determined by methyl thiazolyl tetrazolium (MTT) assay. Data are expressed as mean ± SD *(n* = 3). The blue bar represents the control group corresponding to untreated cells. Statistical significance was measured by one-way analysis of variance followed by Tukey test. * *p* < 0.05 and ** *p* < 0.01 between the control and different concentrations of *N. caudatum* essential oils.

**Figure 2 molecules-27-08193-f002:**
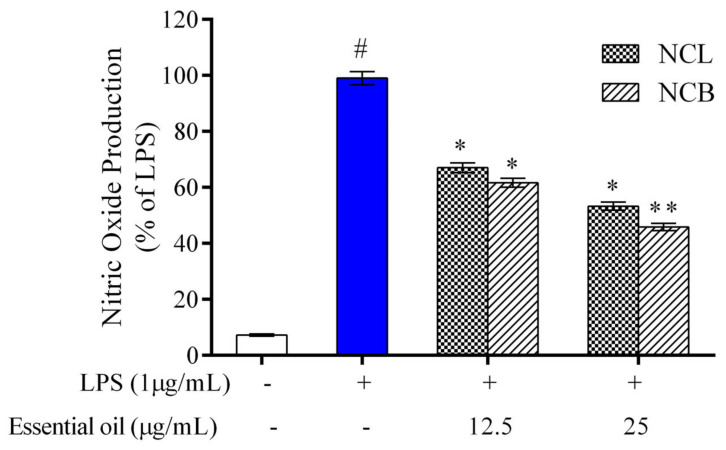
Effect of *N. caudatum* leaf and bark essential oils (12.5 and 25 µg/mL) on NO production in RAW 264.7 cells. The cells were treated with various concentrations of essential oils, followed by treatment with LPS for 24 h. Data are expressed as mean ± SD (*n* = 3). The white bar represents the control group corresponding to untreated cells and the blue bar represents cells treated with LPS (1 µg/mL). Statistical significance was measured by one-way analysis of variance followed by Tukey test. # *p* < 0.05 between the control and LPS-treated cells; * *p* < 0.05 and ** *p* < 0.01 between cells treated with LPS alone (1 µg/mL) and those treated with *N. caudatum* essential oils and LPS.

**Figure 3 molecules-27-08193-f003:**
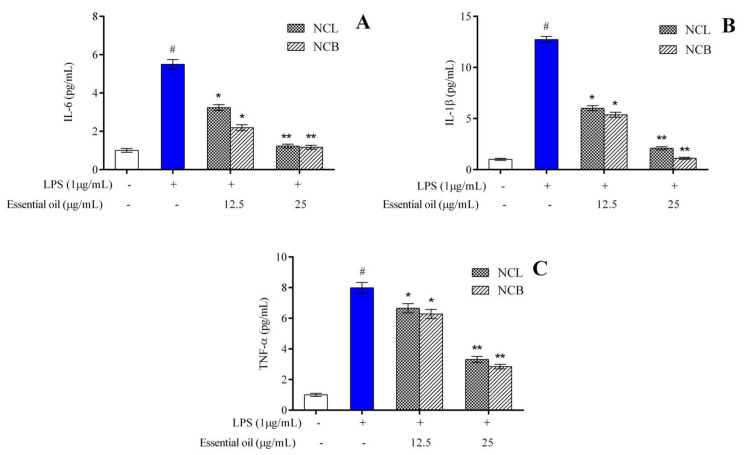
Effect of *N. caudatum* leaf and bark essential oils (12.5 and 25 µg/mL) on LPS-induced proinflammatory cytokine production in RAW 264.7 cells. Cells were treated with *N. caudatum* leaf and bark essential oils (12.5 and 25 µg/mL), followed by 1 μg/mL of LPS for 24 h. After incubation, cell-free supernatants were harvested for estimation of (**A**) interleukin 6 (IL-6), (**B**) interleukin-1β (IL-1β) and (**C**) tumor necrosis factor alpha (TNF-α) via ELISA. Data are expressed as mean ± SD (*n* = 3). The white bar represents the control group corresponding to untreated cells and the blue bar represents cells treated with LPS (1 µg/mL). Statistical significance was measured by one-way analysis of variance followed by Tukey test. # *p* < 0.05 between the control and LPS-treated cells; * *p* < 0.05 and ** *p* < 0.01 between the cells treated with LPS alone (1 µg/mL) and those treated with *N. caudatum* essential oils and LPS.

**Figure 4 molecules-27-08193-f004:**
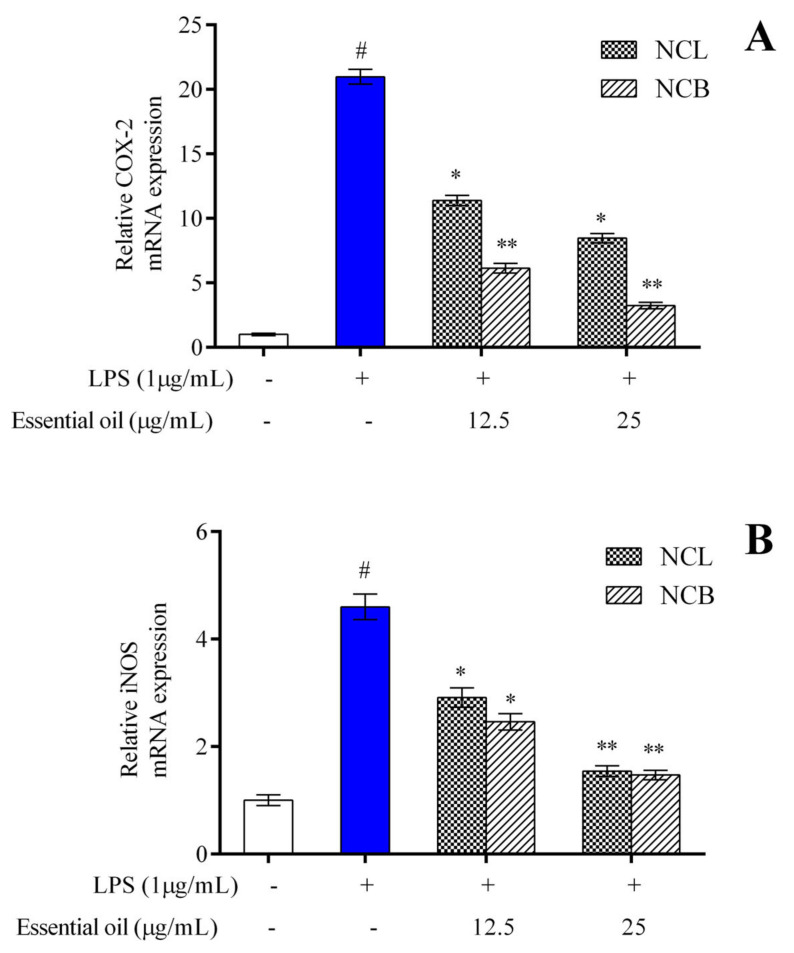
The relative mRNA expression levels of (**A**) COX-2 and (**B**) iNOS determined by qPCR in RAW 264.7 cells treated with *N. caudatum* leaf and bark essential oils (12.5 and 25 µg/mL) followed by treatment with LPS. The white bar represents the control group corresponding to untreated cells and the blue bar represents cells treated with LPS (1 µg/mL). Statistical significance was measured by one-way analysis of variance followed by Tukey test. # *p* < 0.05 between the control and LPS-treated cells; * *p* < 0.05 and ** *p* < 0.01 between the cells treated with LPS alone (1 µg/mL) and those treated with *N. caudatum* essential oils and LPS.

**Figure 5 molecules-27-08193-f005:**
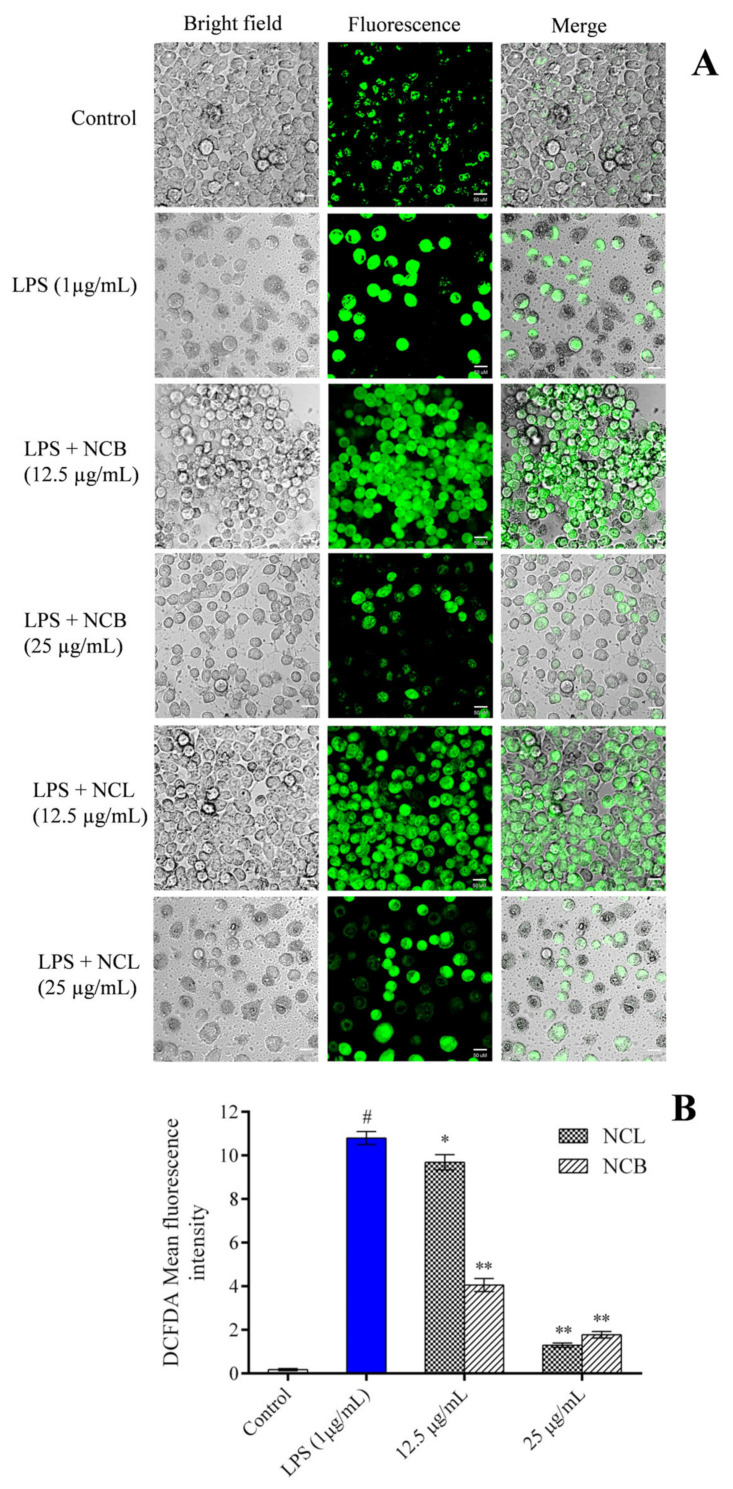
Effect of *N. caudatum* leaf and bark essential oils (12.5 and 25 µg/mL) on LPS-induced ROS production in RAW 264.7 cells. Cells were treated with varying concentrations of essential oil and LPS (1 µg/mL) and incubated for 24 h, followed by treatment with DCFH-DA (5 µM) for 30 min. (**A**) Effects of LPS-induced ROS production by *N. caudatum* leaf and bark essential oil imaged by confocal microscopy (scale bar = 50 uM) using DCFH-DA dye. (**B**) ROS production measured in terms of DCFH-DA mean fluorescence intensity. Data are expressed as mean ± SD (*n* = 3). The white bar represents the control group corresponding to untreated cells and the blue bar represents cells treated with LPS (1 µg/mL). Statistical significance was measured by one-way analysis of variance followed by Tukey test. # *p* < 0.05 between the control and LPS-treated cells; * *p* < 0.05 and ** *p* < 0.01 between the cells treated with LPS alone (1 µg/mL) and those treated with *N. caudatum* essential oils and LPS.

**Figure 6 molecules-27-08193-f006:**
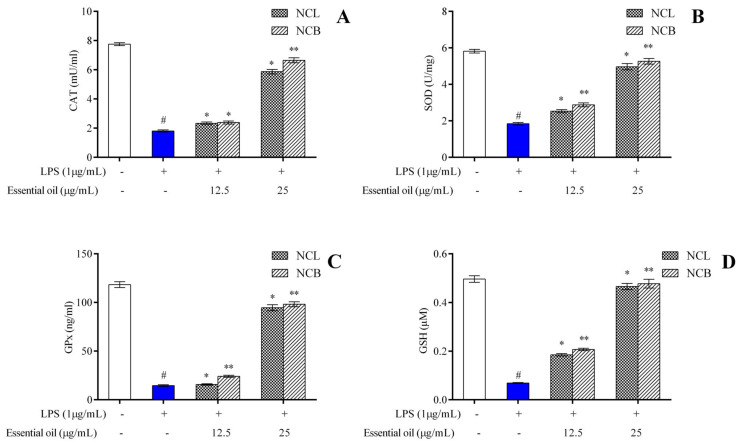
Effect of *N. caudatum* leaf and bark essential oils (12.5 and 25 µg/mL) on the endogenous antioxidant enzymes (**A**) catalase (CAT), (**B**) superoxide dismutase (SOD), (**C**) glutathione peroxidase (GPx) and (**D**) glutathione (GSH) in LPS-induced RAW 264.7 cells. Data are expressed as mean ± SD (*n* = 3). The white bar represents the control group corresponding to untreated cells and the blue bar represents cells treated with LPS (1 µg/mL). Statistical significance was measured by one-way analysis of variance followed by Tukey test. # *p* < 0.05 between the control and LPS-treated cells; * *p* < 0.05 and ** *p* < 0.01 between the cells treated with LPS alone (1 µg/mL) and those treated with *N. caudatum* essential oils and LPS.

**Figure 7 molecules-27-08193-f007:**
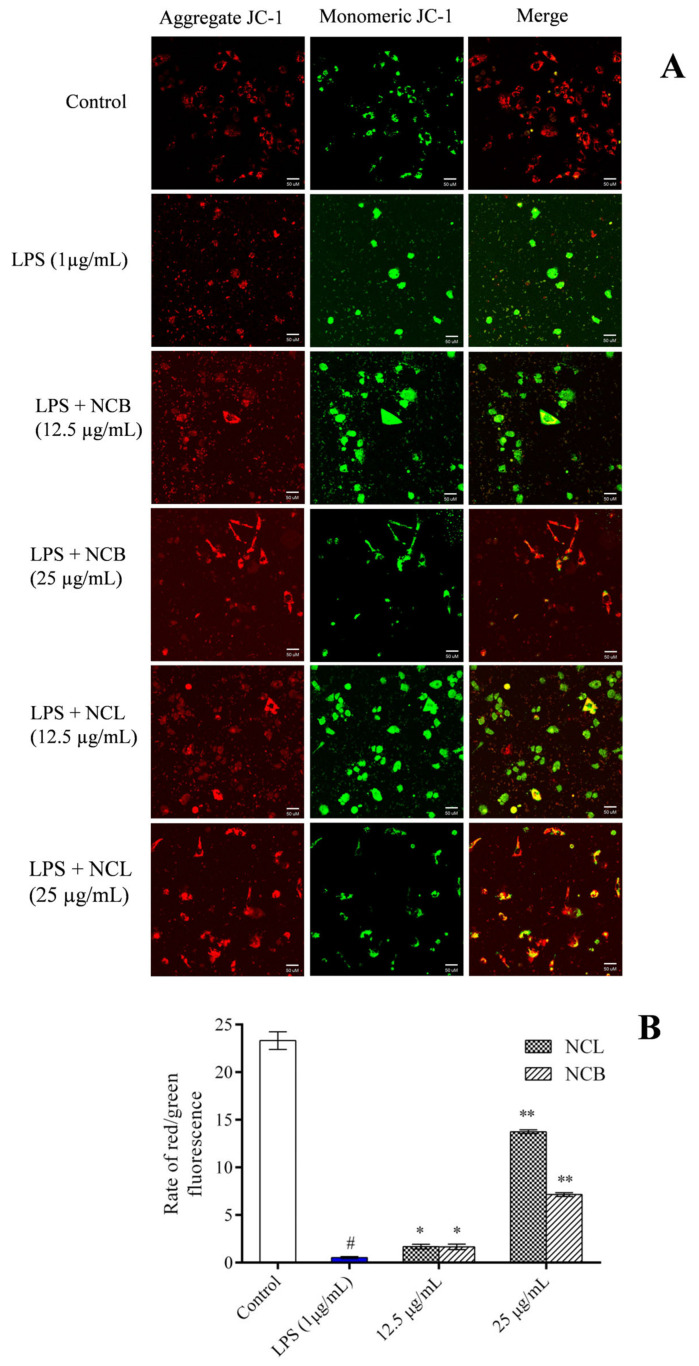
Effect of *N. caudatum* leaf and bark essential oils (12.5 and 25 µg/mL) on LPS-induced changes in mitochondrial potential of RAW 264.7 cells. Cells were treated with various concentrations of essential oils followed by LPS (1 µg/mL) for 24 h. (**A**) Mitochondrial membrane potential was detected via JC-1 staining under a confocal microscope (scale bar = 50 uM); (**B**) quantitative analysis of the ratio of red/green fluorescence in each group measured using Image J software. Data are expressed as mean ± SD (*n* = 3). The white bar represents the control group corresponding to untreated cells and the blue bar represents cells treated with LPS (1 µg/mL). Statistical significance was measured by one-way analysis of variance followed by Tukey test. # *p* < 0.05 between the control and LPS-treated cells; * *p* < 0.05 and ** *p* < 0.01 between the cells treated with LPS alone (1 µg/mL) and those treated with *N. caudatum* essential oils and LPS.

**Figure 8 molecules-27-08193-f008:**
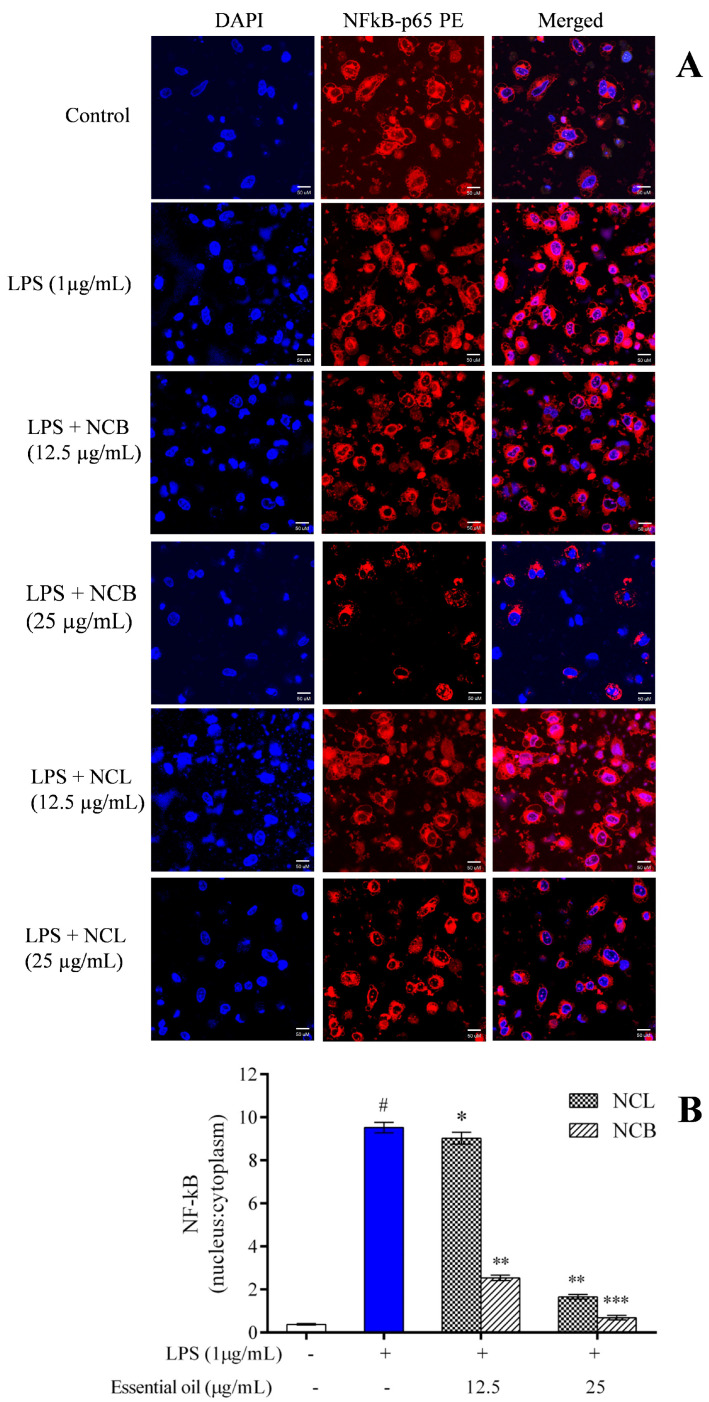
Effect of *N. caudatum* leaf and bark essential oils (12.5 and 25 µg/mL) on LPS-induced nuclear translocation in RAW 264.7 cells. Cells were treated with varying concentrations of essential oil, followed by LPS (µg/mL) for 24 h. (**A**) *In vitro* immunofluorescence assay showing nuclear translocation of the NF-κB-p65 subunit under confocal microscopy (scale bar = 50 uM). (**B**) Quantitative immunofluorescent nuclear: cytoplasmic ratio of the NF-κB-p65 subunit. The white bar represents the control group corresponding to untreated cells and the blue bar represents cells treated with LPS (1 µg/mL). Statistical significance was measured by one-way analysis of variance followed by Tukey test. # *p* < 0.05 between the control and LPS-treated cells; * *p* < 0.05 and ** *p* < 0.01 and *** *p* < 0.001 between the cells treated with LPS alone (1 µg/mL) and those treated with *N. caudatum* essential oils and LPS.

**Table 1 molecules-27-08193-t001:** Chemical constituents of essential oil from the leaf and bark of *Neocinnamomum caudatum*.

**No.**	**Compound**	**RI ^a^**	**RI ^b^**	**Peak Area (%)**	
				**Leaf**	**Bark**
1	α-Thujene	921	930	0.89	0.55
2	α-Pinene	930	939	9.27	10.99
3	α-Fenchene	945	952	0.29	0.58
4	Benzaldehyde	957	960	0.14	-
5	Sabinene	966	975	-	2.30
6	β-Pinene	971	979	45.21	13.11
7	Myrcene	984	990	9.97	7.81
8	α-Phellandrene	1002	1002	-	0.26
9	δ-3-Carene	1003	1011	0.06	0.77
10	α-Terpinene	1012	1017	1.61	2.09
11	p-Cymene	1019	1024	0.29	0.15
12	Limonene	1024	1029	2.18	1.12
13	1,8-Cineole	1028	1031	2.66	-
14	(*Z*)-β-Ocimene	1039	1037	0.24	-
15	(*E*)-β-Ocimene	1052	1050	2.30	3.00
16	γ-Terpinene	1066	1059	1.38	-
17	Terpinolene	1098	1088	1.55	-
18	6-Camphenol	1120	1113	0.40	0.28
19	α-Terpineol	1176	1188	3.95	6.79
20	γ-Terpineol	1190	1199	0.30	1.55
21	Linalool formate	1204	1216	0.08	-
22	endo-Fenchyl acetate	1220	1220	0.20	-
23	Linalool acetate	1277	1257	-	0.39
24	δ-Terpinyl acetate	1327	1317	1.36	0.19
25	δ-Elemene	1352	1338	0.14	-
26	β-Patchoulene	1380	1381	0.97	1.10
27	β-Cubebene	1397	1388	-	0.26
28	β-Elemene	1404	1390	0.11	-
29	(*Z*)-Caryophyllene	1410	1408	2.96	9.30
30	(*E*)-Caryophyllene	1431	1419	0.52	0.12
31	γ-Elemene	1444	1436	0.23	0.66
32	α-Humulene	1446	1454	0.28	-
33	Aromadendrene	1459	1441	0.11	0.22
34	allo-Aromadendrene	1464	1460	0.07	0.76
35	9-epi-(*E*)-Caryophyllene	1470	1466	0.38	-
36	γ-Gurjunene	1477	1477	0.12	0.47
37	γ-Muurolene	1484	1479	0.59	1.24
38	ar-Curcumene	1488	1480	1.10	1.31
39	Germacrene D	1501	1481	0.12	1.51
40	γ-Curcumene	1505	1482	0.41	5.50
41	trans-Calamenene	1512	1522	0.19	0.18
42	δ-Cadinene	1514	1523	0.43	-
43	cis-Calamenene	1521	1529	0.57	0.29
44	γ- Cuprenene	1525	1533	0.08	0.43
45	Germacrene B	1571	1561	1.30	1.07
46	Caryophyllene oxide	1582	1583	-	0.16
47	Humulene epoxide II	1603	1608	-	0.24
48	Guaiol	1630	1600	0.08	3.15
49	10-epi-γ-Eudesmol	1632	1623	0.07	3.49
50	1-epi-Cubenol	1637	1628	-	2.29
51	α-Cadinol	1644	1654	0.17	11.18
52	α-Muurolol	1660	1646	-	0.28
53	α-Bisabolol	1684	1685	0.22	1.79
	Monoterpene hydrocarbons (1–3, 5–12, 14–17)			75.24	42.73
	Monoterpene aldehyde (4)			0.14	-
	Monoterpene ether (13)			2.66	-
	Monoterpene alcohol (18–20)			4.65	8.62
	Monoterpene ester (21–24)			1.64	0.58
	Sesquiterpene hydrocarbon (25–45)			10.68	24.42
	Sesquiterpene ether (46, 47)			-	0.4
	Sesquiterpene alcohol (48–53)			0.54	22.18
	Total identified			95.55	98.93

Data are represented as mean ± SD (*n* = 3). ^a^ Retention indices (RIs) calculated against homologous n-alkane series (C_8_–C_20_) on and Elite-5 MS column. ^b^ RI from the literature.

## Data Availability

The data presented in this work are available in the article.

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
