# Peer review of "Neocinnamomum caudatum Essential Oil Ameliorates Lipopolysaccharide-Induced Inflammation and Oxidative Stress in RAW 264.7 Cells by Inhibiting NF-κB Activation and ROS Production"

_molecules, 2022, doi:10.3390/molecules27238193_

Round 1
Reviewer 1 Report
The paper of Jena and co-authors describes the GCMS characterization and the anti-inflammatory properties of the essential oil extracted from leaves and bark of Neocinnamomun caudatum, a plant used in traditional medicine for the treatment of inflammatory responses.
The topic is not new, since the anti-inflammatory potential of plant essential oils is well known. However, this is the first report describing the chemical composition of essential oil from N. caudatum and could be of interest the chemical and biological characterization of its volatile bioactive components.
Over all the paper is well structured and easy to read. However, there are some concerns regarding the experimental design that I would like the authors better explain.
Specifically, I wondered why the authors ‘pretreat’ the RAW 264.7 macrophages with NCB and NCL, stimulating only after 2 h with LPS as reported in Material and Methods paragraph 3.5. Anti-inflammatory effect should be tested after stimulation with LPS or by adding the extract and LPS at the same time. Otherwise the effects should be described as protective and not curative of inflammation. Other papers the authors cite, i.e. ref. 13 and references therein, do not perform a pre-incubation treatment with anti-inflammatory compounds for cytokine measurements with the same LPS-stimulated cell model. Is there a reason for this different procedure?
Minor concerns:
Results and Discussion:
Paragraph 2.1
-acronyms NCL and NCB should be described the first time they are used in the main text
-ether and aldehyde are terpene derivatives, I suppose, so please use Monoterpene/sesquiterpene ether and Monoterpene aldehyde to identify them both in the text and in Table 1.
-Correct the typo: 5-lipoxygenase
- “The levels of IL-6, IL-1β and TNF-α were significantly elevated from 1.004 to 5.495 pg/ml, 1.001 to 12.730 pg/ml and 1.003 to 7.988 pg/ml, respectively after 24 h treatment with LPS alone” This statement should precede the discussion on the effects of NCB and NCL treatment on pro-inflammatory cytokine levels.
-in Table 1: Z and E must be in italics; add parentheses in (E)-Caryophyllene. I also recommend to better separate the list of individually identified compounds from the extract composition considering terpene subgroups.
Paragraph 2.4- Were LPS and essential oils added to cells at the same time? Was the experimental set-up different from that described for cytokine analysis? Details in the Experimental part are missing
Materials and Methods
Paragraphs 3.6 and 3.8 -Please add details about cell treatment
Author Response
Reviewer 1
Query-1
Over all the paper is well structured and easy to read. However, there are some concerns regarding the experimental design that I would like the authors better explain.
Response-1
Thank you for the insightful suggestion. As per the suggestions we have incorporated the necessary information in the experimental part of our revised manuscript.
Query-2
Specifically, I wondered why the authors ‘pretreat’ the RAW 264.7 macrophages with NCB and NCL, stimulating only after 2 h with LPS as reported in Material and Methods paragraph 3.5. Anti-inflammatory effect should be tested after stimulation with LPS or by adding the extract and LPS at the same time. Otherwise the effects should be described as protective and not curative of inflammation. Other papers the authors cite, i.e. ref. 13 and references therein, do not perform a pre-incubation treatment with anti-inflammatory compounds for cytokine measurements with the same LPS-stimulated cell model. Is there a reason for this different procedure?
Response-2
We apologize for our mistake. We had performed the experiment by adding the sample and LPS at the same time. We have reframed the sentence “The RAW 264.7 cells were incubated with leaf and bark essential oil of N. caudatum (12.5-25 μg/mL), followed by treatment with 1 μg/mL LPS” in material and methods paragraph (3.5) of our revised manuscript.
Query-3
Paragraph 2.1
-acronyms NCL and NCB should be described the first time they are used in the main text
Response-3
As per the reviewer’s suggestion, we have incorporated the acronyms NCL and NCB in the paragraph 2.1 of our revised manuscript.
Query-4
-ether and aldehyde are terpene derivatives, I suppose, so please use Monoterpene/sesquiterpene ether and Monoterpene aldehyde to identify them both in the text and in Table 1.
Response-4
As per the suggestion, we have incorporated monoterpene ether, sesquiterpene ether and monoterpene aldehyde in the text and table 1 of our revised manuscript.
Query-5
-Correct the typo: 5-lipoxygenase
Response-5
As per the reviewer’s suggestion, we have corrected typographical error in the revised manuscript.
Query-6
- “The levels of IL-6, IL-1β and TNF-α were significantly elevated from 1.004 to 5.495 pg/ml, 1.001 to 12.730 pg/ml and 1.003 to 7.988 pg/ml, respectively after 24 h treatment with LPS alone” This statement should precede the discussion on the effects of NCB and NCL treatment on pro-inflammatory cytokine levels.
Response-6.
Thank you for the suggestion. The error has been rectified.
Query-7
-in Table 1: Z and E must be in italics; add parentheses in (E)-Caryophyllene. I also recommend to better separate the list of individually identified compounds from the extract composition considering terpene subgroups.
Response-6
As per the reviewer’s suggestion, we have corrected necessary changes in the revised table. Also the terpene subgroups have been added in the revised manuscript.
Query-7
Paragraph 2.4- Were LPS and essential oils added to cells at the same time? Was the experimental set-up different from that described for cytokine analysis? Details in the Experimental part are missing.
Response-7
As per the reviewer’s suggestion, we have reframed the sentence in paragraph 2.4 and details of the experimental part in the materials and methods (3.6) have been addressed in our revised manuscript.
Query-8
Materials and Methods
Paragraphs 3.6 and 3.8 -Please add details about cell treatment
Response-8
As per the suggestions, we have addressed details of the experiment in the materials and methods section containing paragraph 3.6 and 3.8 of our revised manuscript.
Reviewer 2 Report
This manuscirpt 'Neocinnamomum caudatum essential oil ameliorates lipopolysaccharide-induced inflammation and oxidative stress in RAW 264.7 macrophages by inhibiting NF-κB activation and ROS production' was demonstrated anti-inflammatory and anti-oxidant effects of Neocinnamomum caudatum essential oil. However, there are some questions about the experiment conditions and results. For example, there was no magnification of pictures (Fig. 5). Also, in the bright field images of the LPS group and the LPS+NCL 25 group, the number of cells was much fewer than that of the other groups. In materials and methods section, there are several questions (2 L-glutamine? / The cells (1x10^5 cells/well in 96 well plate, ect.). Moreover, there are many grammatical errors that needs to be fixed (macrophage cells / LPS induced / Figure 5 & 6, ect). Please proofread and review the final draft before submission. Therefore, this manuscript was not suitable for this journal in my opinion.
Author Response
Reviewer 2
Query-1
This manuscirpt 'Neocinnamomum caudatum essential oil ameliorates lipopolysaccharide-induced inflammation and oxidative stress in RAW 264.7 macrophages by inhibiting NF-κB activation and ROS production' was demonstrated anti-inflammatory and anti-oxidant effects of Neocinnamomum caudatum essential oil. However, there are some questions about the experiment conditions and results. For example, there was no magnification of pictures (Fig. 5). Also, in the bright field images of the LPS group and the LPS+NCL 25 group, the number of cells was much fewer than that of the other groups.
Response -1
Thank you for the insightful suggestion. The provided images were captured at the 40x magnification (Scale: 50µm) and the modified images with scale bar have been enclosed in the revised manuscript. Since LPS was toxic to cells it moderately induced stress and apoptosis on cells resulting in shrinkage and blebbing by losing integrity of cell membrane with appearance of low cell density. At a low concentration of NCL (12.5 ug/ml) the protective effect was not much against LPS and the cells we are observing in images are floating or dead cells that have become clumps and debris. However, at higher concentration of NCL (25 ug/ml), protective effect against LPS is enhanced and cells do not form clumps but shows healthy morphology with intact cell membrane.
Query-2
In materials and methods section, there are several questions (2 L-glutamine? / The cells (1x10^5 cells/well in 96 well plate, etc.). Moreover, there are many grammatical errors that needs to be fixed (macrophage cells / LPS induced / Figure 5 & 6, etc). Please proofread and review the final draft before submission.
Response -2
We apologize for our mistakes. As per the reviewer’s suggestion, we have corrected all the mistakes in the materials and methods section.
Reviewer 3 Report
COMMENT:
In the manuscript titled “Neocinnamomum caudatum essential oil ameliorates lipopolysaccharide-induced inflammation and oxidative stress in RAW 264.7 macrophages by inhibiting NF-κB activation and ROS production”, the authors demonstrated that the leaf and bark essential oil of Neocinnamomum caudatum could inhibit the LPS-induced inflammatory and oxidative stress in RAW 264.7 cells, as evidenced by the decrease in the levels of pro-inflammatory cytokines, inflammatory mediators, and intracellular ROS.
Major concerns
1. The rationale of this study has not been clearly addressed. The authors might list more relevant evidence in the Introduction section about the idea of anti-inflammatory effects of the essential oil of Neocinnamomum caudatum or other Neocinnamomum. For example, whether any specific components of the essential oil of Neocinnamomum caudatum or other Neocinnamomum have been shown with anti-inflammatory effects, or any physical or chemical properties are suggested to be involved in the regulation of inflammation or macrophage functions?
2. Authors should describe why ROS inhibition, mitochondrial membrane potential, and NF-κB nucleus translocation in NCB is better than NCL (fig5 and 6).
3. No positive controls (e.g., ROS scavenger NAC) were used in the biological assays.
4. The percentage reduction of various tested markers should have been stated in the results section.
5. The discussion is very weak. Results should be compared to previous reports on the chemical composition and anti-inflammatory activity of Neocinnamomum caudatum or other Neocinnamomum essential oils.
6. In Figure 3: the LPS-stimulated proinflammatory cytokines production is very lower compared to the other reference.
7. Results section for 2.2: please specify the solvent used to prepare different concentrations.
8. In Figure 5: cell density in 25 μg/ml of NCB and NCL was significantly lower compared to 12.5μg/ml of NCB and NCL. However, in figure 1 25 μg/ml of NCB and NCL was no cytotoxicity in RAW 264.7 cells.
9. Please combine figures 5 and 6, figures 8 and 9, as well as figures 10 and 11, they are the same thing.
10. Figures 4: Please provide the results for Western blotting of COX-2 and iNOS
11. Please add the incubation time of NCB, NCL, and LPS in the figure legend
Minor points
- In the materials and methods section (3.3. Cytotoxicity assay and 3.10. Assessment of NF-κB nuclear translocation), please correct the concentration of leaf and bark essential oil.
- Please describe in JC-1 staining, why cells need to be trypsinized
- Results section for 2.5: multiple molecular targets linked to acute and chronic….the following word is cut off
Author Response
Reviewer 3
Query-1
The rationale of this study has not been clearly addressed. The authors might list more relevant evidence in the Introduction section about the idea of anti-inflammatory effects of the essential oil of Neocinnamomum caudatum or other Neocinnamomum. For example, whether any specific components of the essential oil of Neocinnamomum caudatum or other Neocinnamomum have been shown with anti-inflammatory effects, or any physical or chemical properties are suggested to be involved in the regulation of inflammation or macrophage functions?
Response-1
Thanks for the suggestions. In the revised manuscript, we have provided the rationale of the study by justifying the necessity of the work. Being a small genus of 6 species globally, no work has been done on the essential oils of the genus except the chemical composition of Chinese Neocinnamomuim delavayi (Ding et al., 1994). In view of this, we have compared the data on chemical composition and anti-inflammatory activities of the essential of Neocinnamomum caudatum with its closely related species of the genus Cinnamomum at all parts of the revised manuscript.
Besides, as suggested, we have emphasized the anti-inflammatory properties of two prominent components of the essential oil of Neocinnamomum caudatum, i.e. α-pinene and β-pinene to justify the work. Hope this serves the purpose.
Query-2
Authors should describe why ROS inhibition, mitochondrial membrane potential, and NF-κB nucleus translocation in NCB is better than NCL (fig5 and 6).
Response-2
The explanation to the differential anti-inflammatory activity of N. caudatum bark and leaf essential oil has been added as a point (2.9) in the “Results & Discussion” chapter.
However, our explanation for this is given below:
The better anti-inflammatory activity of NCB might be due to the presence of predominant compounds such as α-terpineol and α-cadinol in higher amounts in NCB as compared to NCL. Several reports have shown α-cadinol and α-terpineol to exert significant anti-inflammatory activities (Held et al., 2007; Tung et al., 2010). α-terpineol displayed anti-inflammatory properties by inhibiting cytokine cascade generated by carrageenan and inhibiting the production of nitric oxide in mice. Compound α-cadinol showed potential anti-inflammatory properties by inhibiting NO production in LPS induced macrophages (Tung et al., 2010). Sabinene, a monoterpene hydrocarbon identified in the bark essential oil of N. caudatum was reported to exhibit significant anti-inflammatory effects through the inhibition of pro-inflammatory cytokines such as IL-1β, IL-6, and TNF-α (Yoon et al., 2009). Thus, the significant anti-inflammatory activity of NCB essential oil might be directly related to its compound’s scavenging ability or capacity to inhibit iNOS and COX-2 expression, the enzyme responsible for the release of high amounts of NO, during inflammatory conditions. Additionally, the anti-inflammatory effect of NCB might be due to synergistic interaction of major and minor constituents present in the essential oil.
- Yoon, W.J., Kim, S.S., Oh, T.H., Lee, N.H. and Hyun, C.G., 2009. Cryptomeria japonica essential oil inhibits the growth of drug-resistant skin pathogens and LPS-induced nitric oxide and pro-inflammatory cytokine production. Polish Journal of Microbiology, 58(1), pp.61-68.
- Tung, Y.T., Yen, P.L., Lin, C.Y. and Chang, S.T., 2010. Anti-inflammatory activities of essential oils and their constituents from different provenances of indigenous cinnamon (Cinnamomum osmophloeum) leaves. Pharmaceutical biology, 48(10), pp.1130-1136.
- Rufino, A.T., Ribeiro, M., Sousa, C., Judas, F., Salgueiro, L., Cavaleiro, C. and Mendes, A.F., 2015. Evaluation of the anti-inflammatory, anti-catabolic and pro-anabolic effects of E-caryophyllene, myrcene and limonene in a cell model of osteoarthritis. European journal of pharmacology, 750, pp.141-150.
- Held, S., Schieberle, P. and Somoza, V., 2007. Characterization of α-terpineol as an anti-inflammatory component of orange juice by in vitro studies using oral buccal cells. Journal of agricultural and food chemistry, 55(20), pp.8040-8046.
Query-3
No positive controls (e.g., ROS scavenger NAC) were used in the biological assays.
Response-3
We do agree to the fact that in general NAC, a ROS scavenger is used as a research tool in the field of apoptosis research for investigating the role of ROS in induction of apoptosis. Previously several authors have examined whether or not treated essential oil could reduce the LPS-induced generation of ROS in RAW 264.7 cells using DCF-DA staining devoid of NAC as positive control (Ho et al., 2020; Raka et al., 2022). Considering the fact that our aim was to test the cytoprotective effect of NCB and NCL against ROS, and the results also showed that increase in LPS-stimulated ROS production was markedly attenuated by pre-treatment with NCB and NCL, so co-treatment of NAC with NCB/NCL and LPS was felt not necessary.
- Ho, C.L., Li, L.H., Weng, Y.C., Hua, K.F. and Ju, T.C., 2020. Eucalyptus essential oils inhibit the lipopolysaccharide-induced inflammatory response in RAW264. 7 macrophages through reducing MAPK and NF-κB pathways. BMC complementary medicine and therapies, 20(1), pp.1-11.
- Raka, R.N., Zhiqian, D., Yue, Y., Luchang, Q., Suyeon, P., Junsong, X. and Hua, W., 2022. Pingyin rose essential oil alleviates LPS-Induced inflammation in RAW 264.7 cells via the NF-κB pathway: an integrated in vitro and network pharmacology analysis. BMC Complementary Medicine and Therapies, 22(1), pp.1-16.
Query-4
The percentage reduction of various tested markers should have been stated in the results section.
Response-4
Thank you for the insightful suggestion. As per the suggestion, we have mentioned the percentage reduction of various tested markers in the results section (2.5 and 2.8) of our revised manuscript.
Query-5
The discussion is very weak. Results should be compared to previous reports on the chemical composition and anti-inflammatory activity of Neocinnamomum caudatum or other Neocinnamomum essential oils.
Response-5
Thanks for critically observing this point and giving us suggestions to improve the manuscript.
Kindly note that the genus Neocinnamomum is globally represented by only six species and no work has been done yet on the essential oils of the genus except the chemical composition of Chinese Neocinnamomuim delavayi (Ding et al., 1994). Therefore, it is difficult the compare the data with either N. caudatum or other species of the same genus. As suggested, we have discussed the results on essential oil composition and anti-inflammatory activities with species of a closely related genus Cinnamomum, where several references are available. Such data and references have been inserted at appropriate places in the text.
Query-6
In Figure 3: the LPS-stimulated proinflammatory cytokines production is very lower compared to the other reference.
Response-6
Yes, the production of pro-inflammatory cytokines such as IL-1β, IL-6, and TNF-α in RAW264.7 cells was markedly increased in response to the LPS treatment, whereas pre-treatment with NCB and NCL at both the dose (treated groups) significantly inhibited the secretion of IL-1β, IL-6, and TNF-α. This is due to the fact that NCB and NCL inhibited the expression levels of TNF-α, IL-6, IL-1β] in LPS-activated RAW264.7 macrophages through reducing the activation of NF-κB.
Query-7
Results section for 2.2: please specify the solvent used to prepare different concentrations.
Response-7
The following methodology “Different concentration of essential oil was prepared by dissolving it in DMSO and freshly diluting to the desired concentration with the culture medium. The final concentration of the DMSO was 0.5% (v/v)” has been stated in section 3.3.
Query-8
In Figure 5: cell density in 25 μg/ml of NCB and NCL was significantly lower compared to 12.5 μg/ml of NCB and NCL. However, in figure 1 25 μg/ml of NCB and NCL was no cytotoxicity in RAW 264.7 cells.
Response-8
Cells toxicity was initially determined using MTT assay, wherein it was observed that cells did not exhibit any cytotoxicity till 50 ug/ml. Cell density was same at both the instances i.e at 12.5 and 25 μg/ml. At a low concentration of NCL (12.5 ug/ml) the protective effect was not much against LPS and the cells we are observing in images i.e. Fig 5 are floating or dead cells that have become clumps and debris. However, at higher concentration of NCL (25 ug/ml), protective effect against LPS is enhanced and cells do not form clumps but shows healthy morphology with intact cell membrane.
Query-9
Please combine figures 5 and 6, figures 8 and 9, as well as figures 10 and 11, they are the same thing.
Response-9
As per the reviewer’s suggestion, we have combined figures 5 and 6, figures 8 and 9, as well as figures 10 and 11 in the revised manuscript.
Quer-10
Figures 4: Please provide the results for Western blotting of COX-2 and iNOS
Response-10
Thank you so much for your valuable suggestion. We have not carried out the western blotting analysis of COX-2 and iNOS. The relative mRNA expression level of gene COX-2 and iNOS has been done using quantitative real-time polymerase chain reaction (qRT‒PCR) analysis by taking GADPH as housekeeping gene.
Query-11
Please add the incubation time of NCB, NCL, and LPS in the figure legend
Response-11
We have added the incubation time of NCB, NCL, and LPS in the figure legend of our revised manuscript.
Query-12
Minor points
- In the materials and methods section (3.3. Cytotoxicity assay and 3.10. Assessment of NF-κB nuclear translocation), please correct the concentration of leaf and bark essential oil.
Response-12
Sorry for the mistakes. As per the suggestions, we have corrected the concentration of leaf and bark essential oil in the materials and methods section.
Query-13
Please describe in JC-1 staining, why cells need to be trypsinized.
Response-13
We apologized for the mistake. The cells were not trypsinized. After treatments, cells were washed with PBS and incubated with medium containing JC-1 staining reagent at 37°C for 20 min followed by wash with PBS. Mitochondrial membrane potential was then detected by fluorescence microscopy. The same has been added in the revised manuscript.
Query-14
Results section for 2.5: multiple molecular targets linked to acute and chronic….the following word is cut off
Response -14
As per the reviewer’s suggestion, we have arranged the sentence in results section.
Round 2
Reviewer 2 Report
There are many writing errors or irregulars should be improved. Please proofread and review the final draft before revised manuscript submission.
Author Response
Query-1
There are many writing errors or irregulars should be improved. Please proofread and review the final draft before revised manuscript submission.
Response-1
As per the reviewer’s suggestion, we have rectified the writing errors by taking the help of native English speaking Professors.
Reviewer 3 Report
The current study Neocinnamomum caudatum essential oil ameliorates
lipopolysaccharide-induced inflammation and oxidative stress in RAW 264.7
macrophages by inhibiting NF-κB activation and ROS production. The revised manuscript has improved significantly.
However, the JC-1 staining still has no modification that needs to be corrected in the manuscript.
Author Response
Query-1
However, the JC-1 staining still has no modification that needs to be corrected in the manuscript.
Response-1
As per the reviewer’s suggestion, we have modified the methodology of JC-1 staining in the revised manuscript.